# Treatment with Cenobamate in Adult Patients with Lennox–Gastaut Syndrome: A Case Series

**DOI:** 10.3390/jcm12010129

**Published:** 2022-12-24

**Authors:** Giovanni Falcicchio, Simona Lattanzi, Francesco Negri, Marina de Tommaso, Angela La Neve, Nicola Specchio

**Affiliations:** 1DiBraiN Department, University of Bari ‘Aldo Moro’, 70124 Bari, Italy; 2Department of Experimental and Clinical Medicine, Neurological Clinic, Marche Polytechnic University, 60020 Ancona, Italy; 3Rare and Complex Epilepsy Unit, Department of Neuroscience, Full Member of European Reference Network EpiCARE, Bambino Gesù Children’s Hospital, IRCCS, 00146 Rome, Italy

**Keywords:** epilepsy, cenobamate, Lennox–Gastaut syndrome, epileptic encephalopathy, drug-resistant epilepsy

## Abstract

*Background*. Lennox–Gastaut syndrome (LGS) is a developmental and epileptic encephalopathy (DEE) in which drug resistance to antiepileptic drugs (AEDs) is common. Focal-onset seizures (FOS) are among the seizure types characterizing LGS. Cenobamate (CNB) is a new AED indicated for the treatment of FOS and it has shown promising results in terms of seizure frequency reduction in both clinical trials and real-world experience. To date, the use of CNB in patients with DEEs is limited to Dravet syndrome. *Methods*: This was a retrospective study aimed to determine the 12-month effectiveness and tolerability of CNB in patients with LGS following real-world practice. *Results*: Four patients with LGS receiving CNB treatment were identified. At 12 months from starting CNB, the reduction in baseline seizure frequency ranged from 25 to 74%, with two patients achieving ≥50% seizure reduction. CNB was generally well tolerated and adjustments in doses of concomitant AEDs were required. *Conclusions*: CNB may represent a promising therapeutic option in patients with drug-resistant epilepsy associated with LGS. Further research is needed to confirm this preliminary evidence.

## 1. Introduction

Lennox–Gastaut syndrome (LGS) is a childhood-onset developmental and epileptic encephalopathy (DEE) that usually appears by 8 years of age and shows a peak incidence between 3 and 5 years of age [1]. Although there is no universally accepted definition of LGS due to the heterogeneity of its clinical presentation [2], the condition is conventionally defined by the presence of a triad of features: (1) drug-resistant seizures with variable semiology; (2) bursts of slow spike–wave complexes or generalized paroxysmal fast activity on the interictal electroencephalogram (EEG); (3) cognitive and behavioral impairment [3,4]. Tonic and atonic seizures and atypical absences are the ‘core’ seizures in LGS, even though they may appear later in the course of the disease, or be subtle and not easily recognized (i.e., seizures occurring during sleep or tonic seizures not involving the axial muscles or the entire body) [1,3]. The interictal EEG pattern is not always pathognomonic, and some authors have reported LGS patients without intellectual disability (ID) [5,6]. LGS accounts for 1–2% of all people with epilepsy (PWE) and 1–10% of childhood epilepsies [1,3]. A definite etiology of LGS can be identified in most patients (65–75%), including genetic, structural, and metabolic causes, but the etiology can also be unknown [3]. LGS can evolve from West syndrome in about 20% of patients [7].

The therapeutic management of seizures in LGS is challenging due to high drug resistance to antiepileptic drugs (AEDs) [8]. Valproate (VPA), lamotrigine (LTG), and topiramate (TPM) are first-line pharmacological therapies, followed by rufinamide (RUF), clobazam (CLB), and felbamate (FLB) [3]. Recently, newer AEDs such as cannabidiol (CBD) and fenfluramine (FLM) [9] have shown efficacy in LGS patients. Despite the increasing number of available therapeutic options, long-term outcomes remain poor due to both drug-resistant seizures and impairment in psychosocial functioning [3].

Focal seizures with or without bilateral tonic–clonic evolution can occur in LGS and sometimes precede the ‘core’ seizure types [10]. Cenobamate (CNB) is the most recently approved AED for the adjunctive treatment of focal-onset seizures with or without secondary generalization in adult PWE whodid not achieve adequate seizure control despite a history of treatment with at least two AEDs [11]. Dual, complementary mechanisms of action are thought to contribute to the anti-seizure activity of CNB: it enhances the inactivation of voltage-gated sodium channels by preferentially inhibiting persistent rather than transient currents, and acts as a positive allosteric modulator of GABA_A_ receptor binding at non-benzodiazepine sites [11]. In preclinical studies, CNB has shown a broad spectrum of anti-seizure activity in various rodent seizure models, i.e., electrical, chemical, and kindling models of both focal and generalized seizures [12]. In clinical studies, a promising seizure freedom rate of about 20% was observed in adult patients with uncontrolled focal epilepsy [13], and the effectiveness of CNB has also been reported in pediatric cohorts of patients with refractory seizures [14,15]. To date, data on the efficacy and tolerability of CNB in patients with LGS are lacking. In this study, we reported preliminary evidence about the use of CNB as an adjunctive treatment for LGS in clinical practice.

## 2. Methods

### 2.1. Study Participants

This was a retrospective study with the aim to determine the efficacy and tolerability of add-on CNB in patients with LGS. Patients with LGS were identified among the patients treated with CNB within the Early Access Program at the Epilepsy Center of Bari University Hospital (Policlinico of Bari), Italy, between 2020 and 2022. Patients were followed up every three months through clinical visits, as is routine practice at the center when a new AED is added. Inclusion criteria were age ≥ 18 years, focal seizures, and drug-resistant epilepsy. Patients were considered drug-resistant if they continued to have seizures despite at least two adequate trials of tolerated and appropriately chosen and used schedules of AEDs (whether as monotherapies or in combination) according to the current international definition [16]. Demographic-, seizure- and treatment-related data were obtained from medical records. The efficacy outcomes were reduction in the monthly frequency of all seizures compared to baseline (4 weeks before adding CNB) and seizure response (reduction ≥ 50% from baseline seizure frequency) at 12 months; 12-month seizure freedom (defined as the occurrence of no seizure since at least the previous visit) was also considered. Percentage change in seizure frequency at 12 months for each patient was calculated as ([seizure frequency per 28 days] − [seizure frequency at baseline])/[seizure frequency at baseline] × 100. Tolerability outcomes included the type, severity and duration of adverse effects (AEs).

### 2.2. Statistical Analysis

Continuous variables are summarized as mean ± standard deviation (SD) or median [interquartile range], and categorical variables are presented as the number (%) of patients. The normal distribution of continuous variables was checked with the Shapiro–Wilk test. Sample size and study power were not calculated due to the descriptive nature of the study, which was a retrospective case series. The STATA/IC 13.1 statistical package (StataCorp LP, College Station, TX, USA) was used to perform statistical analysis.

## 3. Results

Four patients with LGS were identified. The mean age of patients was 32.3 (5.0) years, and all were male. The age at epilepsy onset ranged from 3 months to 6 years and the mean duration of epilepsy was 29.8 (6.0) years. Three patients had severe ID and one patient had moderate ID. In two cases, LGS developed from West syndrome. All patients were drug-resistant, and they had a mean number of 14.8 (4.3) prior AEDs. Only one patient implanted vagus nerve stimulation (VNS). None of them underwent epilepsy surgery or tried ketogenic diet. The mean number of concomitant AEDs was 2.5 (1.0) at the time of the introduction of CNB, and the median baseline monthly seizure frequency was 45.8 [20.0–322.3]. The clinical characteristics of the included patients are summarized in Table 1.

Patient 1 had LGS with a combined structural and genetic etiology (bilateral temporo-occipital cortical dysplasia plus the c.3871C > T mutation in LAMC3 gene). He was 29 years old and had moderate ID. Seizures with variable semiology (myoclonic, tonic, generalized tonic–clonic (GTCSs), and focal) occurring with high frequency (several times a day) were reported since six years of age. He had no history of status epilepticus (SE). Generalized fronto-temporal spikes and polyspikes were identified on the most recent EEG. Despite previous therapeutic attempts with 14 AEDs and VNS, he continued to experience multiple daily seizures. Cenobamate 12.5 mg/day was added to VPA 1500 mg/day, zonisamide (ZNS) 400 mg/day, and carbamazepine (CBZ) 1400 mg/day. At 12 months of follow up, by which time he was taking CNB 250 mg/day, a seizure reduction of 25% was recorded. Due to the occurrence of ataxia, dizziness and vomiting, CBZ was gradually down-titrated and eventually discontinued, and the AEs completely resolved. Anti-seizure therapy at 12 months was CNB 250 mg/day, VPA 1500 mg/day, and ZNS 400 mg/day.

Patient 2 was a 36-year-old man with severe ID and a history of treatment with nine different AEDs. At three months of age, he started to experience epileptic spasms and was diagnosed with West syndrome. Other seizure types appeared later. At baseline, he had high seizure frequency (multiple daily), experiencing absences, tonic seizures, focal seizures, and GTCSs. Brain MRI revealed cerebellar atrophy, and the c.3523_3524delCT mutation in the COL18A1 gene was identified through whole-exome sequencing. No history of SE was reported. EEG before CNB administration showed generalized fronto-temporal spikes and sharp waves. He was taking VPA 1550 mg/day when CNB 12.5 mg/day was added. At the last follow up, the patient was taking CNB 300 mg/day and seizure frequency had decreased by 69% compared to baseline. No AEs were reported. The same dose of VPA was used throughout the 12 months.

Patient 3 was diagnosed with West syndrome at the age of five months. He continued to present myoclonic seizures and other seizure types (including focal seizures) despite prior treatment with 18 different AEDs. Brain MRI revealed right frontal focal cortical dysplasia, and the most recent EEG showed diffuse theta activity. He was 27 years old at the time of CNB initiation. At baseline, he was taking VPA 800 mg/day, CLB 30 mg/day, and ZNS 300 mg/day. During CNB titration, moderate somnolence appeared and progressive reduction in CLB was attempted, leading to resolution of the AE. The pharmacological burden was further reduced with progressive down-titration, and then the withdrawal of VPA. At the 12-month follow up, the CNB dose was 200 mg/day and a 33% reduction in seizure frequency was recorded; concomitant AEDs were CLB 10 mg/day and ZNS 300 mg/day.

Patient 4 was a 37-year-old man with LGS carrying the hemizygous tPro555LeufsTer52(c.1664del) variant in the IQSEC2 gene. Delivered by forceps, he presented ID and spastic tetraparesis. A percutaneous endoscopic gastrostomy tube was placed due to severe dysphagia. At three years of age, he started to present atonic and focal seizures, followed by GTCSs and tonic seizures at 15 years of age. No history of SE was reported. Brain MRI showed bilateral frontal micropolygyria. The most recent EEG showed bilateral frontal theta activity. Despite previous treatment attempts with 18 different AEDs, he continued to present GTCSs (2–3/year) and atonic and focal seizures (both multiple times a day). Cenobamate was started at 12.5 mg/day, added on to VPA 1800 mg/day, PB 75 mg/day, and RUF 1800 mg/day. During CNB titration, he reported moderate somnolence, which resolved completely after the reduction in PB dose. At 12 months, when he was taking CNB 200 mg/day, a seizure frequency reduction of 74% compared to baseline was recorded; concomitant AEDs were VPA 1800 mg/day, PB 50 mg/day, and RUF 1800 mg/day.

Details about the efficacy and tolerability outcomes of the four patients are summarized in Table 2.

## 4. Discussion

In this small cohort of LGS patients, CNB showed a good efficacy and tolerability profile. All patients showed a reduction in baseline seizure frequency during treatment with CNB, and two were responders, i.e., presented a seizure reduction ≥50% compared to baseline. No seizure freedom was reached in any case, even though some patients were seizure-free for some days during the titration period. This might be explained by the physiological fluctuation of seizures in the natural history of epilepsy, or by the fact that the occurrence of AEs necessitated adjustments of concomitant AEDs during the CNB treatment.

The titration schedule of CNB was always applied unless AEs occurred, in which case the titration was temporarily suspended to allowdose adjustment of concomitant AEDs. Once AEs remitted, CNB titration was resumed.

The main AEs were central nervous system (CNS)-related. As expected, patients taking PB and CLB showed moderate somnolence, possibly due to pharmacokinetic interactions involving CYP2C19 [11]. A dose reduction in the latter drugs was required. In the only patient taking an AED acting as a sodium channel blocker, the presence of dizziness and vomiting, even in presence of a reduction in CBZ blood levels, might be explained by a pharmacodynamic synergism between the two drugs. In general, the AEs resolved in a short space of time. In Patient 1, the AEs lasted longer, possibly because the CBZ dose was progressively reduced in parallel with the slow up-titration of CNB. In the end, the CBZ was discontinued and a final dose of CNB 200 mg/day was reached with the complete resolution of AEs.

No serious AEs occurred. In particular, no patient showed hypersensitivity adverse reactions such as drug rash with eosinophilia and systemic symptoms (DRESS). Hepatic transaminases and potassium [17] levels remained within the normal ranges throughout the 12-month follow up in all the patients. Electrocardiographic monitoring did not show any alterations.

Although no objective assessment method was adopted, caregivers reported a global improvement in the behavior and sleep of the patients with severe ID.

Importantly, no patients discontinued CNB, and they were all being treated with this drug at the 12-month follow up.

Limited evidence exists so far about the effectiveness of CNB in patients with LGS. Even though a cohort of patients with LGS was included in the study by Connor et al. [18], no specific information about this subgroup of PWE was provided and discussed. In a recent paper, the use of CNB in a small cohort of patients with another DEE, i.e., Dravet syndrome, was examined, and significant seizure reduction was reported [19].

In brief, our preliminary data showed that:AEs were less likely to occur in patients with a lower pharmacological burden;Patients taking PB and CLB experienced moderate-to-severe somnolence, completely resolved after the reduced dose of these AEDs;The only patient taking a sodium channel blocker AED (CBZ) presented severe ataxia with vomiting and dizziness, and CNS-related AEs resolved progressively with slow down-titration and the withdrawal of CBZ;The only patient taking RUF did not show ECG variations, such as QT and QTc shortening.

## 5. Limitations of the Study

This study provides preliminary and anecdotal evidence about the use of CNB in patients with LGS. The retrospective collection of data and the inclusion of patients admitted to one single center may have introduced potential sources of bias. The small sample size did not allow us to perform comparison between groups, correlation analyses, or inferential statistics to identify predictors of efficacy and tolerability. The response to CNB was evaluated by estimating the reduction in the frequency of all seizure types, and no sub-analyses according to the different types of seizures were performed. Data about seizure frequency were obtained through the interpretation of seizure diaries and clinical records of follow-up visits, and no other objective assessments of drug efficacy were performed. In this regard, although self-reported calendar diaries still represent the main source to evaluate drug efficacy in clinical studies, different issues may affect the validity of these tools, and future directions to improve the reliability of seizure count may include electroencephalographic biomarkers and automatic seizure detection linked to electronic diaries [20]. The lack of data about the plasma levels of CNB prevented the performance of dose–response analyses and the exploration of interactions between drugs. The absence of a control group treated with an alternative option meant that we could not compare the effectiveness of CNB and other AEDs. The current study therefore did not allow us to draw definitive conclusion about the effectiveness of CNB in patients with LGS, but could stimulate hypotheses to be evaluated through additional studies in larger and prospective cohorts of patients.

## 6. Conclusions

Data on the use of CNB in LGS are still lacking. The present real-world experience suggests that CNB may exert a beneficial effect in terms of seizure frequency reduction in this population of patients with difficult-to-treat seizures. The high pharmacological burden likely contributes to the occurrence of AEs and a proactive dose adjustment of concomitant AEDs based on the knowledge of both pharmacokinetic and pharmacodynamic interactions that can occur may be useful to improve tolerability. The association with sodium channel blocker AEDs seemed to give a less favorable outcome in terms of AEs. Caution is advised when prescribing CNB to patients taking this class of AEDs. Further research is needed to fully characterize the potentialities and clinical relevance of CNB as a therapeutic option for patients with LGS.

## Figures and Tables

**Table 1 jcm-12-00129-t001:** Baseline characteristics of the patients.

	Patient 1	Patient 2	Patient 3	Patient 4
Age (years)	29	36	27	37
Sex	M	M	M	M
Epilepsy etiology	Structural and genetic	Genetic	Structural	Structural and genetic
Genetic mutation *	c.3871C > T (LAMC3 gene)	c.3523_3524delCT (COL18A1 gene)	Investigation ongoing	c.1664del (IQSEC2 gene)
Intellectual disability	Yes	Yes	Yes	Yes
Intellectual disability (severity)	Moderate	Severe	Severe	Severe
Age at epilepsy onset	6 years	3 months	5 months	3 years
Epilepsy duration (years)	23	35.7	26.5	34
Development from West syndrome	No	Yes	Yes	No
Seizure types				
Absences	-	+	-	-
Atonic seizures	-	-	+	+
Myoclonic seizures	+	+	+	-
Tonic seizures	+	+	+	+
Tonic–clonic seizures	+	-	+	+
Status epilepticus	-	-	-	-
Focal seizures	+	+	+	+
EEG abnormalities before initiation of CNB	Generalized fronto-temporal spikes and polyspikes	Generalized fronto-temporal spikes and sharp waves	Diffuse theta activity	Bilateral frontal theta activity
Brain MRI	Bilateral temporo-occipital cortical dysplasia	Cerebellar atrophy	Right frontal cortical dysplasia	Bilateral frontal micropolygyria
Epilepsy surgery	No	No	No	No
VNS	Yes	No	No	No
AEDs prior to CNB (number)	14	9	18	18
AEDs prior to CNB	CBZ, CLB, GVG, LCM, LEV, LTG, NTZ, OXC, PB, RUF, TPM, VPA, ZNS, corticosteroids	ACTH, BRV, CBZ, CLB, PB, RUF, TPM, VPA, immunoglobulins	ACTH, BRV, CLB, CNZ, ESL, ESM, GVG, LCM, LEV, LTG, NTZ, PB, PER, RUF, TPM, VPA, ZNS, corticosteroids	CBZ, CLB, CLZ, ESM, FLB, GBP, GVG, LEV, LTG, OXC, PB, PER, PHT, RUF, TGB, TPM, VPA, ZNS
AEDs at initiation of CNB (number)	3	1	3	3
AEDs at initiation of CNB	CBZ, VPA, ZNS	VPA	CLB, VPA, ZNS	PB, RUF, VPA
Baseline monthly seizure frequency	26.5	65	13.5	579.5

ACTH = adrenocorticotropic hormone; BRV = brivaracetam; CBZ = carbamazepine; CLB = clobazam; CNB = cenobamate; CNZ = clonazepam; ESL = eslicarbazepine; ESM = etosuccimide; FLB = felbamate; GBP = gabapentin; GVG = vigabatrin; LCM = lacosamide; LEV = levetiracetam; LTG = lamotrigine; NTZ = nitrazepam; OXC = oxcarbazepine; PB = phenobarbital; PER = perampanel; PHT = phenytoin; TGB = tiagabine; TPM = topiramate; RUF = rufinamide; VPA = valproate; ZNS = zonisamide. AED = antiepileptic drug; EEG = electroencephalogram; MRI = magnetic resonance imaging; VNS = vagus nerve stimulation. * Genetic mutations identified through whole-exome sequencing.

**Table 2 jcm-12-00129-t002:** Efficacy and tolerability outcomes at 12 months from starting cenobamate.

	Patient 1	Patient 2	Patient 3	Patient 4
CNB dose (mg/day)	250	300	200	200
Concomitant AEDs (number)	2	1	2	3
Seizure frequency reduction	25%	69%	33%	74%
Seizure response	No	Yes	No	Yes
Seizure freedom	No	No	No	No
Adverse event (yes/no)	Yes	No	Yes	Yes
Type of adverse event	Ataxia, dizziness and vomiting	-	Somnolence	Somnolence
Intensity of adverse event	Moderate/severe	-	Moderate	Moderate
Duration of adverse event	9 months	-	2 weeks	2 months

AED = antiepileptic drug; CNB = cenobamate.

## Data Availability

Data are available on reasonable request from the corresponding author.

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
