# Peer review of "Treatment with Cenobamate in Adult Patients with Lennox–Gastaut Syndrome: A Case Series"

_jcm, 2022, doi:10.3390/jcm12010129_

Round 1
Reviewer 1 Report
Firstly, I would like to congratulate the authors on their interesting and relevant work.
However, there are some points that need to be addressed:
1) The title should contain the study design type.
2) In the abstract, the authors should describe what is the main research question and the study design type. Also, it should contain a brief description of how their study relates to previous research.
3) Preferably key-words should be standardized to increase the visibility of the article. I advise authors to choose their key-words from MeSH portal. Please see this site for more information: https://www.ncbi.nlm.nih.gov/mesh/
4) I advise authors to change the term “anti-seizure medications (ASMs)” to “Antiepileptic drugs (AEDs)” along the text because this is the preferred terminology by ILAE.
5) What was the main research question?
6) Why specifically was CNB chosen instead of other AED?
7) How was the seizure frequency assessed?
8) What was the frequency of following-up assessments of the patients?
9) Was video-EEG performed?
10) What was the criteria used for the inclusion of patients in the study?
11) The authors should provide more information on the drug-resistance status of the patients. Where the patients classified as having DRE during the entire observation period? Which definitions of DRE they used?
12) How was the sample size calculated?
13) What was the power of the study?
14) What was the distribution of data?
15) The authors should describe statistical analyzes in the methods section. Considering the time of following-up of the study, it would be interesting if authors could further elaborate on the statistical analyzes. Were any type of comparative analyzes performed? Any correlation studies? Confidence intervals and p-values should be reported. Were any objective assessments of efficacy of the drug performed?
16) Taking into account the small sample size and the subjectivity of the evaluation of the drug efficacy, it is difficult to draw any conclusions and this should be further explained in the discussion.
17) How were genetic mutations assessed in the patients?
18) Why there were not any controls evaluated in the study?
19) Considering CNB is a newly approved drug, this study categorizes as an experimental study. As such, authors should provide the registration number and name of the trial registry. Also, an appropriate description of the study design should be included in the methods section
20) Table 1 should be improved. For example, there are abbreviations in the table not explained in the legend.
21) Were serum levels of CNB assessed? Were there any associations of them with adverse events?
22) The discussion section should further address several other limitations of the study.
23) The conclusion section should be improved and no new information should be included on it.
Reviewer 2 Report
This article, submitted from Italy, discusses the clinical efficacy of the new drug cenobamate (CNB) in adults with Lennox-Gastaut syndrome (LGS). The subjects were four cases of LGS and CNB was administered for 12 months. Seizure reduction was observed in all four cases after treatment initiation. There were no serious adverse reactions, although dose adjustments were necessary for the concomitant medications already administered. Therefore, the continued accumulation of clinical efficacy in this case may be useful.
Overall, there are no obvious problems with the proofreading of the paper. The description of the clinical information is also qualified. The reviewers offer some comments below and the authors are invited to examine the responses.
Minor 1: If there are characteristic changes or improvements in the EEG findings in the four cases, is it possible to present an EEG?
Minor 2: Age-dependent epilepsy syndromes, or DEE, but please more describe precisely. West syndrome, Ohtahara syndrome, early myoclonic encephalopathy and infantile epilepsy with migratory focal seizures, which are age-dependent in the neonatal to early childhood period and which may later progress to LGS.
Minor 3: The causes of LGS are described in the Introduction. More precisely, the underlying conditions that cause it include cortical dysplasia, hypoxic ischaemic encephalopathy, post-traumatic brain injury, brain tumours, metabolic abnormalities, chromosomal abnormalities, congenital malformation syndromes and genetic abnormalities. However, no common pathologies have been found. The authors suggest some modifications.
Mnor 4: There is mention of epilepsy surgery and VNS, but what about ketogenic diet therapy as a treatment?
Minor 5: In refractory epilepsy with three or more drugs, the number of cases showing benefit from the addition of a new drug is very small, reported to be around 3%. In this paper, two out of four cases show a clear benefit. Why not emphasise this point further?
Minor 6: What were the blood concentration levels of CNB?
Minor 7: What are the key points about interactions between CNB and other drugs, if any findings?
Minor 8: What is the dosage form of CNB? Is it in tablet, powder or liquid form? Were the four patients on oral medication? Or were they administered by tube?
Mnor 9: Finally, do you have any recommendations for epileptic drugs that can be combined with CNB? Please let us know if you have any reasoning of your own.
Best regards,
Dr. reviewer
Round 2
Reviewer 1 Report
The authors have replied adequately to all of the reviewer's comments.
Author Response
Thank you very much!